# Isolation and Genome-Based Characterization of Biocontrol Potential of *Bacillus siamensis* YB-1631 against Wheat Crown Rot Caused by *Fusarium pseudograminearum*

**DOI:** 10.3390/jof9050547

**Published:** 2023-05-09

**Authors:** Qianqian Dong, Qingxiang Liu, Paul H. Goodwin, Xiaoxu Deng, Wen Xu, Mingcong Xia, Jie Zhang, Runhong Sun, Chao Wu, Qi Wang, Kun Wu, Lirong Yang

**Affiliations:** 1College of Life Sciences, Henan Agricultural University, Zhengzhou 450046, China; dongqianqian@hnagri.org.cn (Q.D.);; 2Institute of Plant Protection Research, Henan Academy of Agricultural Sciences, Henan Agricultural Microbiology Innovation Center, Zhengzhou 450002, China; 3School of Environmental Sciences, University of Guelph, Guelph, ON N1G 2W1, Canada; 4College of Plant Protection, China Agricultural University, Beijing 100193, China

**Keywords:** Fusarium crown rot, *Bacillus siamensi*s, *Fusarium pseudograminearum*, biocontrol, genome, growth promotion

## Abstract

Fusarium crown rot (FCR) caused by *Fusarium pseudograminearum* is one of the most serious soil-borne diseases of wheat. Among 58 bacterial isolates from the rhizosphere soil of winter wheat seedlings, strain YB-1631 was found to have the highest in vitro antagonism to *F. pseudograminearum* growth. LB cell-free culture filtrates inhibited mycelial growth and conidia germination of *F. pseudograminearum* by 84.14% and 92.23%, respectively. The culture filtrate caused distortion and disruption of the cells. Using a face-to-face plate assay, volatile substances produced by YB-1631 inhibited *F. pseudograminearum* growth by 68.16%. In the greenhouse, YB-1631 reduced the incidence of FCR on wheat seedlings by 84.02% and increased root and shoot fresh weights by 20.94% and 9.63%, respectively. YB-1631 was identified as *Bacillus siamensis* based on the *gyrB* sequence and average nucleotide identity of the complete genome. The complete genome was 4,090,312 bp with 4357 genes and 45.92% GC content. In the genome, genes were identified for root colonization, including those for chemotaxis and biofilm production, genes for plant growth promotion, including those for phytohormones and nutrient assimilation, and genes for biocontrol activity, including those for siderophores, extracellular hydrolase, volatiles, nonribosomal peptides, polyketide antibiotics, and elicitors of induced systemic resistance. In vitro production of siderophore, β-1, 3-glucanase, amylase, protease, cellulase, phosphorus solubilization, and indole acetic acid were detected. *Bacillus siamensis* YB-1631 appears to have significant potential in promoting wheat growth and controlling wheat FCR caused by *F. pseudograminearum*.

## 1. Introduction

Fusarium crown rot (FCR) of wheat, caused by *Fusarium pseudograminearum*, can result in the rotting of seeds, seedlings, roots, crowns, subcrowns, and lower stems, inhibiting the flow of water and nutrients [1]. It has become one of the most important diseases of wheat, primarily in arid and semi-arid regions of the world. Yield losses caused by FCR are approximately 10% but can be over 30% when conditions favor the disease, such as drought or high humidity, temperature, and soil organic matter [2]. Although *F. pseudograminearum* only colonizes wheat roots up to the second stem node, the mycotoxin deoxynivalenol (DON) also can be detected in the head, creating a serious threat to the health of humans and animals [3].

Biocontrol agents (BCAs) are a very promising alternative to chemical pesticides for plant disease control. Many plant-growth-promoting bacteria can produce abundant antifungal secondary metabolites and hydrolytic enzymes with strong antagonistic activities against plant pathogenic fungi, as well as trigger-induced systemic resistance (ISR) [4]. Among bacterial BCAs, *Bacillus* species are considered to be one of the most valuable for field application, producing a wide range of different antimicrobial compounds. They utilize 5–8% of their genomes for secondary metabolite synthesis, and the major antimicrobial metabolites are lipopeptides, non-ribosomally synthesized peptides, polyketides, bacteriocins, and siderophores [5].

*Bacillus siamensis* was first described in salted crab in Thailand in 2010 [6]. It is widely distributed in the environment, such as in the beltfish gastrointestinal tract [7], rhizosphere soil [8], fermented foods, and the interior of plant tissues [9]. It has been shown to have plant growth promotion activity for a wide range of plants, including wheat [10], sweet pepper [11], tomato [12], and Chinese cabbage [13]. It has also been shown to have the potential to control a range of pathogens by direct antimicrobial activity, such as against charcoal rot fungus [14], post-harvest fruit rot fungi [15], and Fusarium head blight fungus [16]. Additionally, it has the potential to control diseases by inducing host resistance, such as against post-harvest anthracnose of mango [17], fruit rot and wilt of sweet pepper [18], and brown spot of tobacco [8]. 

Sequencing of the genome of *B. siamensis* KCTC13613 revealed that it has genes for promoting root colonization, including those for chemotaxis (methyl-accepting chemotaxis proteins and CheV system), biofilm formation (polyglutamic acid, biofilm-surface layer protein, extracellular polysaccharides, biofilm matrix protein), and motility (flagellar and swarming motility proteins). There were also genes for plant growth promotion, including those for phytohormone synthesis (auxin and gibberellic acid) and nutritional assimilation (iron, nitrogen, phosphorus, and potassium). Genes were also found for production of antimicrobial volatile organic compounds (VOCs) (acetoin and 2,3- butanediol), nonribosomal peptide (surfactin and fengycin), polyketides (bacillaene and difficidin), and a hybrid nonribosomal peptide/polyketide (iturin). In addition, there were genes for the production of compounds related to induced systemic resistance (ISR), such as VOCs (acetoin and 2,3- butanediol) and microbe-associated molecular patterns (MAMPs) (peptidoglycan, flagellin, and elongation factor Tu). Many of these genes have also been found in the genomes of other *B. siamensis* strains [7,19]. Thus, *B. siamensis* strains can have a wide range of plant growth promotion and biocontrol traits.

In this study, strain YB-1631 was isolated from winter wheat rhizosphere soil and was selected from 58 isolates based on its strongest in vitro antagonistic ability against the *F. pseudograminearum*. It was identified as *B. siamensis*, and its antifungal activity in vitro and in vivo and growth promotion ability on wheat seedlings were tested. The genome of strain YB-1631 was sequenced and assembled, and genes related to root colonization, plant growth promotion, antimicrobial activity, and ability to induce host disease resistance were analyzed. This is the first report on the biocontrol of wheat FCR with *B. siamensis*.

## 2. Materials and Methods

### 2.1. Isolation of Strain YB-1631 and In Vitro Antifungal Assays

Two rhizosphere soil samples from 50 wheat seedlings each were collected from two fields of winter wheat in Maqiao, Henan, China. A total of 1 g of soil was dilution plated on LB agar and incubated at 37 °C. Pure cultures from bacterial colonies with different colony characteristics were maintained on Luria-Bertani (LB) agar.

For bacterial colony mycelial growth inhibition, one 5 mm agar plug of *F. pseudograminearum* WZ-8A was inoculated onto the center of a PDA plate, and then the bacterial isolates were inoculated 2.5 cm from the fungal plug and incubated at 28 °C. The distance of the zone between the fungal and bacterial colonies was measured after 5 days [20]. 

For bacterial cell-free culture filtrate mycelial growth inhibition, strain YB-1631 was grown in LB broth overnight at 37 °C and 180 rpm and then inoculated into LB, Landy [21], or mineral salt medium (MSM) [22] broth. After incubation at 37 °C and 180 rpm for 72 h, the broths were centrifuged at 17,226× *g* for 20 min, and the supernatant passed through a 0.22 µm sterile filter. The filter-sterilized extracts were mixed into potato dextrose agar (PDA) at 50 °C to make 5%, 10%, 15%, and 20% PDA + LB, Landy, or MSM extract. The control was PDA alone. Each plate was inoculated with a 5 mm *F. pseudograminearum* agar plug and then incubated at 28 °C. The growth diameter of *F. pseudograminearum* was measured at 5 days. The percent inhibition rate of mycelial growth was calculated: (colony diameter on PDA alone–colony diameter on extract + PDA)/colony diameter on PDA alone × 100 [23]. 

For bacterial cell-free culture filtrate conidial germination inhibition, *F. pseudograminearum* was grown in 100 mL sodium carboxymethyl cellulose medium (CMC) broth [24] at 28 °C and 180 rpm for 5 days. The broth was filtered through Miracloth (475855-1R, Calbiochem, Darmstadt, Germany), centrifuged at 1914× *g* for 10 min, and then the conidia were manually diluted to 1 × 10^6^ spores/mL in 66% LB bacterial extract + yeast extract peptone dextrose medium (YEPD) broth. The control was YEPD broth alone. Cultures were incubated at 28 °C and 180 rpm for 12h, and 100 conidia were examined with three replicates (observed at 400 times magnification under a ZEISS Imager A2 microscope). Conidia were considered to have germinated if the germ tube exceeded half the length of the conidia. The percent inhibition rate of conidial germination was calculated: (conidial germination rate with YEPD alone − conidial germination rate with LB extract + YEPD)/conidial germination rate with YEPD alone × 100 [25]. 

For bacterial volatile mycelial growth inhibition, YB-1631 was grown overnight in LB broth as described above, and then 80 µL of 1 × 10^8^ bacterial cells/mL were spread onto LB agar. A 5 mm *F. pseudograminearum* agar plug was inoculated onto the center of the PDA, then the LB agar and PDA plates without lids were attached face-to-face with parafilm. The control was face-to-face attached plates of non-inoculated LB agar and PDA with *F. pseudograminearum.* Plates were incubated at 28 °C for 5 days. The mycelial growth diameter of *F. pseudograminearum* was measured, and the percent inhibition rate of mycelial growth was calculated as described above [26]. Each experiment was repeated three times.

### 2.2. Biocontrol of FCR and Growth Promotion of Wheat by YB-1631

Surface sterilized wheat seeds (Zhengmai 1860) were germinated for 2 days to obtain primary root emergence [27] and then treated with 10^8^ bacterial cells/mL YB-1631 or sdH_2_O for 4 h at 28 °C. *F. pseudograminearum* was grown in wheat grain sand culture for 1 week at 28 °C to produce soil inoculum [20]. Germinated seeds (21 per pot) were planted in 300 g of sterile soil with or without 6 g *F. pseudograminearum* soil inoculum (2%, *W/W*), and the pots were placed in a greenhouse with 12 h light/12 h dark at 25 °C. At 21 days after sowing, the seedlings were evaluated using a 0 to 7 Fusarium crown rot (FCR) symptom rating scale, with class 0 having no symptoms, classes 1 to 6 having different ranges of necrotic lesion areas, and with plant death as class 7 [28]. Fusarium crown rot incidence (FCRI) was calculated: FCRI (%) = (total number of seedlings − number of non-infected seedlings class value)/total number of seedlings × 100 [29]. Disease severity index (DSI) was calculated: DSI (%) = [Σ(class value × number of seedlings in each class value)/(total number of seedlings × 7) × 100 [29]. Relative control effect (RCE) was calculated: RCE (%) = (DSI of sdH_2_O treatment − DSI of YB-1631 treatment)/DSI of sdH_2_O treatment × 100 [20]. Shoot height, root length and fresh weight, and total fresh weight were also determined at 21 days post-sowing. The experiment was repeated five times.

### 2.3. Wheat Disease Resistance Enzyme Activities

At 20 days post-sowing [20], 0.1 g of stem tissue from YB-1631 treated or non-treated wheat seedlings grown in soil containing or not containing *F. pseudograminearum* were taken and stored at −80 °C. Activities of peroxidase (POD, EC 1.11.1.7), superoxide dismutase (SOD, EC 1.15.1.1), catalase (CAT, EC 1.11.1.6), polyphenol oxidase (PPO, EC 1.10.3.1), lipoxygenase (LOX), phenylalanine ammonia lyase (PAL, EC 4.315), and malondialdehyde (MDA) content were determined by their respective detection kits (Cat. Nos. BC0095, BC0175, BC0205, BC0195, BC0325, BC0215, and BC0025, Beijing Solarbio Science Technology Co., Beijing, China).

### 2.4. Plant Growth Promotion and Biocontrol Traits of YB-1631

Amylase, protease, β-glucanase [30], siderophore, cellulase, and phosphorus solubilization activities were determined from zone diameters on corresponding agar plates as previously reported [20]. IAA production was determined in NB containing L-tryptophan and treated with Salkowski reagent [20,31].

### 2.5. Identification of YB-1631

Gram staining was performed as per Xu et al. [32]. For SEM, YB-1631 was scraped off LB agar at 16 h at 37 °C, fixed with 2.5% glutaraldehyde, and examined with a Hitachi SU8100 SEM microscope (Hitachi, Tokyo, Japan). Genomic DNA of YB-1631 was extracted using the MiniBEST Bacterial Genomic DNA Extraction Kit Ver. 3.0 (Takara, Beijing, China). A 1200 bp *gyrB* sequence was amplified from the YB-1631 genomic DNA using primers UP-1S and UP-2Sr [33] and sequenced (Tsingke Biotechnology, Beijing, China). The *gyrB* sequence was used as a query with blastn against the NCBI nr database. In addition, 16 other *gyrB* sequences were selected for the construction of a tree using the neighbor-joining method with 1000 bootstrap iterations [20,34].

The genome of YB-1631 was sequenced and assembled as described below. The complete genomes of 10 strains of *Bacillus* were downloaded from the EzBioCloud (http://www.ezbiocloud.net, accessed on 12 December 2022), and pairwise comparisons of Average Nucleotide Identity (ANI) of the genomes were calculated using the JSpeciesWS Online Service [35]. The ANI values were used to draw a heatmap using TBtools software [36]. Strains with ANI values higher than 95–96% are defined as the same bacterial species [37,38,39].

### 2.6. Genome Sequencing, Assembly, and Annotation

YB-1631 was grown in LB broth for 12 h at 37 °C and 180 rpm, and genomic DNA was extracted as described above. Sequencing of 250 bp paired-end reads was conducted at Frasergen (Wuhan, Hubei, China) using MGIseq2000 (Mgi Tech Co., Shenzhen, China) and 900 bp single end reads with PacBio Sequel II (Pacific Biosciences, Menlo Park, CA, USA). The genome was assembled using pb_assembly_microbial in Smrtlink (10.1.0; Pacific Biosciences) and Pilon (1.24) [40]. The coverage depth was analyzed with Minimap (2.15-r905) [41] to align three sets of HiFi reads to the assembled genome. Coding regions were identified with Glimmer (3.02) [42] and annotated using blastp in Diamond (2.0.9.147) [42] against the Genbank nr, COG/KOG, GO, SwissProt, KEGG basic databases (cut-off E value 1 × 10^−5^). Genes for the synthesis of secondary metabolites were identified with antiSMASH (https://antismash.secondarymetabolites.org, accessed on 6 December 2022). Genes for extracellular hydrolases were identified using blastp with queries obtained from extracellular hydrolase predicted proteins from *B. velezensis* FZB42 [43]. Genes annotated for γ-PGA synthesis and other plant growth promotion traits were selected based on the literature [20,43,44,45]. tRNAs were predicted with tRNAscan-SE (2.0.9) [46], rRNAs were predicted with RNAmmer (1.2) [47], and sRNAs were predicted from the Rfam database using Cmscan in Infernal (1.1.4) [48]. Clustered regularly interspaced short palindromic repeat sequences (CRISRRs), genomic islands, and prophages were predicted with MinCED (0.4.2) [49], IslandPath-1.0.6 [50], and PhiSpy-4.2.19 [51], respectively. A genome circle map was generated with Circos (0.69–9) [52].

### 2.7. Statistical Analysis

Statistical analyses were performed using IBM SPSS Statistics 21 using one-way ANOVA with Tukey’s test (*p* < 0.05).

## 3. Results 

### 3.1. Isolation and In Vitro Antifungal Activity of YB-1631 

Of 58 bacterial strains isolated from wheat rhizosphere soil, 50 showed some in vitro antifungal activity against *F. pseudograminearum* on PDA plates (Table 1). The largest zone of growth inhibition between the pathogen and soil bacteria was with strain Mq202003-16 at 5.07 mm, which was significantly greater than the other bacterial strains, except for Mq202003-10, Mq202003-23, Mq202003-30, Mq202003-45, and Mq202003-58. Mq202003-16 was deposited in the Henan Academy of Agricultural Sciences soil microbial culture collection as YB-1631.

YB-1631 cell-free culture filtrate from LB, Landy, and MSM media all inhibited the mycelial growth of *F. pseudograminearum* (Figure 1A). The 20% LB culture filtrate resulted in the highest antifungal activity at 84.14% (Figure 1B). Conidial germination of *F. pseudograminearum* was inhibited with 66% LB culture filtrate of YB-1631 by 92.23% compared to the control (Figure 1C). The cell walls of conidia and germ tubes with control treatment appeared intact and without deformation (Figure 2A), while conidia treated with cell-free culture filtrates showed lysis and deformations, and the germ tubes showed swelling, protoplast condensation, shortening of septum intervals, crumpling, and lysis (Figure 2B, arrows a to g). Antifungal volatiles of YB-1631 were detected with a face-to-face plate assay showing inhibition of *F. pseudograminearum* growth by 68.16% (Figure 1C).

### 3.2. Identification of YB-1631

YB-1631 colonies were slimy, translucent, convex, smooth, and pale yellow on LB agar at 12 h at 37 °C (Figure 3A). Under light microscopy, YB-1631 appeared rod-shaped and was Gram-positive (Figure 3B). Under SEM, the rod-shaped cells had an average size of 3.22 μm × 0.72 μm and were surrounded by an extracellular matrix (Figure 3C). 

Based on *gyrB* sequences of eight *Bacillus* species, YB-1631 clustered with *Bacillus siamensis* B28, *B. siamensis* SRCM100169, and *B. siamensis* KCTC13613 (Figure 4A). Based on genome comparisons with five other *Bacillus* species, YB-1631 also had the highest ANI value of 98.69% with *B siamensis* SRCM100169, followed by 98.47% with *B. siamensis* KCTC13613. Both ANI values are higher than the 95–96% identity threshold for defining bacterial species. In contrast, ANI values with other *Bacillus* species ranged from 72.02% to 93.91% (Figure 4B). Thus, YB-1631 was identified as a strain of *B. siamensis.*

### 3.3. B. siamensis YB-1631 In Vivo Biocontrol and Plant Growth Promotion Activities

Wheat seedlings treated or not treated (control) with YB-1631 showed no symptoms of FCR, whereas wheat seedlings inoculated with *F. pseudograminearum* showed FCR symptoms of stunting, black or brown leaf sheaths, and root rot (Figure 5A,B). Seedlings with YB-1631 treatment plus *F. pseudograminearum* inoculation appeared similar to the control and YB-1631 treatment alone. For YB-1631 treatment plus *F. pseudograminearum* inoculation, DI and DSI for FCR were significantly less than with *F. pseudograminearum* alone, with an RCE of 83.23% (Table 2).

YB-1631 treatment significantly increased the activities of all the defense enzyme activities tested and significantly decreased MDA content compared to control non-inoculated plants (Table 3). Inoculation with *F. pseudograminearum* also significantly increased the defense enzyme activities, but it decreased MDA content compared to the control. YB-1631 treatment plus *F. pseudograminearum* inoculation significantly increased the activities of all the defense enzymes relative to the control and other treatments, except for SOD relative to *F. pseudograminearum* alone. MDA content with YB-1631 treatment plus *F. pseudograminearum* inoculation was significantly reduced relative to all other treatments except for YB-1631 alone.

At 21 days post-sowing with YB-1631 treatment, wheat root length, plant height, root fresh weight, and total fresh weight were significantly higher than the non-treated control (Table 2). In contrast, inoculation of *F. pseudograminerum* significantly decreased all those parameters compared to the control. YB-1631 treatment plus *F. pseudograminearum* inoculation resulted in root length, root fresh weight, and total fresh weight significantly greater than the control, while plant height was not significantly different from the control. All those parameters, however, were significantly greater than *F. pseudograminearum* inoculation alone.

### 3.4. Genome Assembly and Annotation of B. siamensis YB-1631

A total of 276,759,717 bp were obtained with PacBio Sequel II sequencing, and 2,205,483,000 bp were obtained using MGIseq2000 sequencing. This resulted in a 67.38× coverage depth of the assembled genome. The assembled genome consisted of one chromosome of 4,090,312 bp and 45.92% GC content (Figure 6). The PacBio Sequel II, MGIseq2000, and YB-1631 assembled genome sequences were deposited at NCBI with the accession numbers SRR22106234, SRR22077931, and CP110268, respectively.

The genome of YB-1631 was predicted to contain 4357 protein-coding genes, 86 tRNAs, 27 rRNAs and 89 other potential RNAs, 6 CRISPR sequences, 10 gene islands, and 9 prophages (Appendix A). Using the NCBI nr, COG, GO, SwissProt, and KEGG databases, annotations were obtained for 96.26%, 82.81%, 72.85%, 51.16%, and 71.47% of the predicted protein sequences, respectively, giving a total annotation of 96.33% (Appendix A). 

### 3.5. Genes and Activities Potentially Related to FCR Biocontrol, Plant Growth Promotion, and the Interaction of B. siamensis YB-1631 with Plants

Ten gene clusters of secondary metabolite synthesis were predicted using the antiSMASH database (Table 4 and Appendix A). For nonribosomal peptide synthesis gene clusters, there were 20 genes in cluster 1 for surfactin, 14 genes in cluster 2 for fengycin, and 13 genes in cluster 3 for bacillibactin. For transAT polyketide synthesis, there were 14 genes in cluster 4 for bacillaene and 15 genes in cluster 5 for difficidin. Cluster 6 for polyketide-like synthase had 2 genes for butirosin A, clusters 7 and 8 for terpene synthesis had 4 genes in cluster 7 for squalene/phytoene and 3 genes in cluster 8 for squalene-hopene, and clusters 9 and 10 for nonribosomal peptide + polyketide synthase had 10 genes in cluster 9 for bacillomycin D and 5 genes in cluster 10 for locillomycin.

In addition, YB-1631 was likely to produce some small antimicrobial compounds because 5 genes for amylocyclin and one each gene for uberolysin/carnocyclin family circular bacteriocin, and antimicrobial peptide Lci antibacterial were found in the YB-1631 genome. However, YB-1631 likely cannot produce bacilysin because only one transporter gene for bacilysin was found, and it cannot produce macrolactin, plantazolicin, and aurantinin for the lack of synthetic genes.

The YB-1631 genome also contains 51 extracellular hydrolase genes (Appendix A). Corresponding to the two genes for β-glucanase, one gene for amylase, one gene for cellulase, and ten genes for protease, the enzymatic activity of β-glucanase, amylase, cellulase, and protease was detected in vitro from YB-1631 (Figure 7B,E,F,G).

The YB-1631 genome also contains many putative plant growth promotion-related genes (Appendix A). There were 4 indole-3-acetic acid synthesis genes, 2 cytokinin synthesis genes, 5 acetoin and butanediol synthesis genes, 4 nitrogen-related genes, 10 phosphorus-related genes, 3 potassium assimilation and transport genes, and 41 iron assimilation and transport genes, which included 22 siderophore genes. Indole-3-acetic acid production, phosphorus solubilization, and siderophore activity were also detected in vitro (Figure 7A,C,D).

For genes potentially related to plant–microbe interaction in the YB-1631 genome, there were 11 genes related to bacterial chemotaxis, 28 genes for flagellar assembly and motor proteins, 4 genes for swarming motility, 62 genes for quorum sensing, and 54 genes for biofilm formation, including 5 genes for the synthesis of the secreted polymer, γ-PGA, 4 genes for bacterial target molecules of the general plant immune response, 9 genes for lipopolysaccharide biosynthesis, and 9 genes for teichuronic acid/lipopolysaccharide biosynthesis (Appendix A). For genes involved in the breakdown, transport, and utilization of plant-derived substrates, there were 11 genes related to cellulose and hemicellulose, 1 gene related to chitin and chitosan, 2 genes related to protein and peptide, and 20 genes related to opine (Appendix A). Finally, in the category of response to environmental stress, there were 2 genes for oxidative stress, 3 genes for nucleic acid-binding protein, and 1 gene for nucleic acid modification (Appendix A).

## 4. Discussion 

The wheat rhizosphere bacterial isolate, YB-1631, was identified as *Bacillus siamensis* based on *gyrB* sequences, ANI values, and morphology. *Bacillus siamensis* is a common rhizosphere soil bacterium, and many isolates have shown biocontrol potential against plant pathogens, such as *B. siamensis* S3 from sweet potato soil with activity against *Pestalotiopsis versicolor* [53], *B. siamensis* GL-02 from maize soil with activity against *F. graminearum* [54], and *B. siamensis* H30-3 from Chinese cabbage soil with activity against *Pectobacterium carotovorum* subsp. *carotovorum* PCC21 [13]. Additionally, many *B. siamensis* isolates from rhizosphere soil have shown plant growth promotion activity, such as *B. siamensis* H30-3 that significantly increased biomass of Chinese cabbage [13], *B. siamensis* RGM 2529 from forest soil that increased biomass of tomato [55], and *B. siamensis* Pbbb1 from bean soil that also increased biomass of tomato [12]. In addition to rhizosphere soil, *B. siamensis* isolates have been reported in fermented and pickled foods [16,56,57], healthy plant roots, leaves, and seeds [9,58,59,60], animal intestines [61], marine sediments [19], sewage [62], and insect hives and nests [63].

*Bacillus siamensis* YB-1631 showed the strongest in vitro antagonistic activity against *F. pseudograminearum* among the isolates tested in this study. It also showed strong in vivo activity against FCR, with a biocontrol efficacy in the greenhouse at 83%, which was higher than that reported for *B. velezensis* YB-185 at 66% [25], *B. subtili*s YB-15 at 81% [20], and *B. halotolerans* QTH8 at 62% [64]. Wheat defense enzyme activities and MDA content were also altered by *B. siamensis* YB-1631, indicating that treatment with this isolate may have resulted in induced systemic resistance of wheat against *F. pseudograminearum* and reduced the oxidative damage of wheat seedlings caused by *F. pseudograminearum* [65]. There are many reports of *Bacillus* isolates inducing resistance against plant pathogens in a variety of crops [66]. In wheat, some examples of induced systemic resistance due to *Bacillus* isolates are resistance to *Septoria tritici* [67], *Blumeria graminis* f. sp. *tritici* [68], and *Fusarium graminearum* [69]. 

Inhibition of *F. pseudograminearum* by *B. siamensis* YB-1631 may have been due to secreted antifungal compounds detected in cell-free culture filtrates. Inhibition of mycelial growth with 5–20% *B. siamensis* YB-1631 LB culture filtrate ranged from 45 to 84%. In comparison, 10–50% NB culture filtrate of *B. velezensis* YB-185 inhibited *F. pseudograminearum* by 44 to 79% [25], 10–50% LB culture filtrate of *B.amyloliquefaciens* Y1 inhibited *F. graminearum* by 34 to 45% [70], and 1–10% LB culture filtrate of *B. velezensis* BM21 inhibited *F. graminearum* by 42 to 76% [71]. Thus, *B. siamensis* YB-1631 LB culture filtrate has similar levels of inhibitory activity against *F. pseudograminearum* and *F. graminearum* as those from other *Bacillus* spp. 

The antifungal compounds identified in cell-free culture filtrates of *B. siamensis* isolates include the lipopeptides, surfactin [22,72,73,74], fengycin [9,19], and iturin A [22,74,75], bacillomycin F [22,74], bacillomycin D [73], and the catechol-based siderophore, bacillibactin [19]. The YB-1631 cell-free culture filtrate resulted in ruptured, swollen, and vacuolated hyphae of *F. pseudograminearum*. In comparison, surfactin causes outgrowths, distortions, protrusions, and swelling of hyphae [76], fengycin causes hyphal cell-surface shrinking [77], iturin A causes swelling and reduced mycelial cytoplasmic content [78], and bacillomycin D results in sunken, lumpy, and wrinkled hyphae [7]. Thus, the damage to the hyphae due to *B. siamensis* YB-1631 cell-free culture filtrate is consistent with the effects of a number of antifungal compounds reported from *B. siamensis* isolates.

One factor that can affect the production of antimicrobial compounds in cell-free culture filtrates from *Bacillus* species is the type of culture medium. All the media tested in this study have been reported to result in the production of culture filtrates containing antifungal compounds, such as LB with *B. siamensis* S3, YC-9, and WB1 against *Pestalotiopsis versicolor, Fusarium oxysporum,* and *Colletotrichum acutatum* [53,79,80], Landy with *B. siamensis* BRBac21-1 against *Macrophomina phaseolina* and *Fusarium oxysporum* f. sp. *udum* [81], and MSM with *B. siamensis* JFL15 against *Magnaporthe grisea*, *Rhizoctonia solani*, *Colletotrichum gloeosporioides,* and *Fusarium graminearum* [7,16,22]. Although previous studies have used different media, this is the first report comparing multiple media, and cell-free culture filtrate of *B. siamensis* YB-1631 grown in LB showed the highest antifungal activity. LB media may be the best as it had the lowest carbon-to-nitrogen ratio (C/N) among the four media tested. Fonseca et al. [82] reported that the lowest C/N ratio (3 versus 9 or 15) tested in a mineral media resulted in the greatest biomass and highest production of antifungal lipopeptides by *B. subtilis* YRE207. However, other media nutrients are important, such as phosphorus and metal ions, which were critical for the production of antifungal lipopeptides by *Bacillus* sp. BH072 [83]. Further research is needed to establish the relationship between specific nutrients and antimicrobial production by *B. siamensis* YB-1631. 

Inhibition of *F. pseudograminearum* by *B. siamensis* YB-1631 may also have come from secreted volatile antifungal compounds that can diffuse through the soil and inhibit conidia germination and mycelial growth [70,84] and volatiles from *B. siamensis* YB-1631 grown on LB agar inhibited mycelial growth of *F. pseudograminearum* by 68%. In comparison, *B. siamensis* G-3 grown on NB agar inhibited raspberry post-harvest diseases caused by *Botrytis cinerea* and *Rhizopus stolonifer* by 89%, and the main antifungal compounds detected were 2, 6-di-tert-butyl-4-methylphenol and 2,4-di-tert-butylphenol [15]. Volatiles of *B. siamensis* LZ88 grown on NB agar inhibited the mycelial growth of *A. alternata* by 78%, and the active compounds were 2-methylbutanoic acid and 3-methylbutanoic acid [8,75]. Volatiles of *B. siamensis* N-1 grown on NB agar inhibited mycelial growth of *C. gloeosporioides*, *P. microcystis*, and *F. incarnatum* by 87%, 75% and 43%, respectively, and the active compounds were N-11-undecene, 3-methyl-1-butanol, 2-nonanone, 1,3,5,7-cyclooctatetraene, and phenol [72]. Thus, a variety of volatile compounds may be responsible and similar to levels of inhibition of fungal growth by volatiles of other *B. saimensis* isolates have been higher than that found in this study.

Sequencing the genome of *B. siamensis* YB-1631 revealed a number of gene clusters potentially related to antibiotic production. Gene clusters were found for the production of lipopeptides, polyketides, dipeptides, cyclic peptides, cationic peptides, and aminoglycosides antibiotics. In comparison, there were similar or identical numbers of genes for such compounds in the genomes of four other *B.siamensis* isolates reported to have antimicrobial, biocontrol, and/or plant growth promotion activities, *B. siamensis* KCTC 13613T [85], *B. siamensis* SCSIO 05746 [19], *B. siamensis* JFL15 [22], and *B. siamensis* RGM 2529 [55] (Appendix A). However, there were a few exceptions. Among those isolates, there were many fewer genes for difficidin in *B. siamensis* RGM 2529 and many more genes for bacilysin in *B. siamensis* SCSIO 05746. However, it appears that except for aurantinin, macrolactin, and plantazolicin, which were limited to the isolates, *B. siamensis* RGM 2529, *B. siamensis* SCSIO 05746, and *B. siamensis* JFL15, respectively, the *B. siamensis* YB-1631 genome appears to encode all the antibiotics found in those other biocontrol *B. siamensis* isolates.

Extracellular hydrolases produced with PGPRs have been reported to be involved in antifungal and plant growth-promotion processes, such as glucanases, amylase, protease, and chitosanases to degrade fungal cell walls [86,87,88], and siderophores and phosphatases to promote plant absorption of minerals [87,88,89]. Genes were also identified in the *B. siamensis* YB-1631 genome for a variety of extracellular hydrolases. These included glucanases, lipases, protease, and chitosanases that have been found in a number of *Bacillus* isolates, such as *Bacillus* sp.739, *B. subtilis* NPU 001, *B. circulans* MH-K1, *B. subtilis* CW14, and *B. subtilis* strain J9 that show antifungal activity against *Bipolaris sorokiniana*, *Fusarium oxysporum*, *Penicillium expansum*, *Aspergillus awamori*, and *Aspergillus ochraceus*, *Fusarium graminearum,* and *Fusarium avenaceum* [87,88,89,90]. Comparing the *B. siamensis* YB-1631 genome to the genomes of *B. siamensis* KCTC 13613T, *B. siamensis* SCSIO 05746, *B. siamensis* JFL15, and *B. siamensis* RGM 2529 revealed that there were similar numbers of genes for extracellular hydrolases with a few exceptions (Appendix A). Only the *B. siamensis* YB-1631 and *B. siamensis* KCTC 13613T genomes contained the ribonuclease gene *yokF*, only the genome of *B. siamensis* KCTC 13613T was missing the ribonuclease gene *yhcR*, only the genome of *B. siamensis* RGM 2529 was missing the ribonuclease gene *yobL,* and only the *B. siamensis* YB-1631 genome contained the protease gene *blaSE*. Thus, there were a few differences between the strains in their extracellular hydrolase genes, and further research, such as gene knockouts and gene expression patterns, is needed to determine if that affects their biocontrol activity. 

In addition to FCR control, *B. siamensis* YB-1631 also significantly increased root and shoot growth in healthy wheat seedlings. Root length and root fresh weight without *F. pseudograminearum* stress was increased by 37% and 21%, respectively, by *B. siamensis* YB-1631. In comparison, *B. siamensis* Pbbb1 in healthy tomato increased root length and root fresh weight by 50% and 92%, respectively [12], and *B. siamensis* MH559649 in healthy wheat cv. NARC-2009 increased root length by 15% [10]. Thus, *B. siamensis* YB-1631 was within the range of increased root growth produced by other *B. siamensis* isolates. Plant height and total plant fresh weight was increased by 14% and 10% by *B. siamensis* YB-1631. Similar results were found for other *B. siamensis* isolates, such as *B. siamensis* Pbbb1 in healthy tomato, which increased plant height and total fresh weight by 15% and 65%, respectively [10], and *B. siamensis* MH559649 in healthy wheat cv. NARC-2009, which increased plant height and total fresh weight by 13% and 15%, respectively [12].

*B. siamensis* YB-1631 also significantly increased root and shoot growth of wheat seedlings with *F. pseudograminearum* infection. This could be due to a combination of plant growth promotion and disease biocontrol. Root length and root fresh weight with *F. pseudograminearum* infection was increased by 88% and 116% by *B. siamensis* YB-1631, which was much greater than with healthy plants. This was greater than *B. subtilis* YB-15 in wheat seedlings with *F. pseudograminearum* infection, which increased root length and root fresh weight by 9% and 68%, respectively [20], and somewhat similar to *B. subtilis* YB-04 in cucumber seedlings with *Fusarium oxysporum* f.sp. *cucumerinum* infection, which increased root length and fresh weight by 46% and 230%. Plant height and total fresh weight with *F. pseudograminearum* infection were increased by 88% and 89% by *B. siamensis* YB-1631. In comparison, *B. subtilis* YB-15 increased plant height and total fresh weight by 26% and 14%, respectively, in wheat seedlings with *F. pseudograminearum* infection [20], and *B. subtilis* YB-04 increased plant height and total fresh weight by 190% and 27%, respectively, in cucumber seedlings with *F. oxysporum* f.sp. *cucumerinum* infection [91].

The *B. siamensis* YB-1631 genome contained a number of genes potentially related to plant growth promotion. There were genes for the synthesis of the phytohormones IAA and cytokinin. IAA is the main auxin in plants regulating cell division and differentiation [92], and cytokinin can act synergistically with IAA to promote cell differentiation and growth [93]. There were also genes for nutrient acquisition, such as phosphate, nitrate/nitrite, potassium, and iron assimilation found in many PGPRs [94], and genes for the production of the polyamines putrescine and spermidine. Polyamines increase plant growth, development, and tolerance to abiotic stress by acting as signaling molecules inside plant cells [95]. Finally, genes for the VOCs acetoin and 2,3- butanediol were identified in the *B. siamensis* YB-1631 and the four other *B. siamensis* genomes. Acetoin and 2,3- butanediol improve plant growth dependent on cytokinin activation [96]. The same number of those genes were also found in the genomes of *B. siamensis* KCTC 13613T, *B. siamensis* SCSIO 05746, *B. siamensis* JFL15, and *B. siamensis* RGM 2529, except for the *B. siamensis* RGM 2529 genome, which lacked the iron assimilation genes *efeM* and *efeU* (Appendix A). Thus, *B. siamensis* YB-1631 and these other *B. siamensis* isolates have a nearly identical array of genes for these potential plant growth promotion traits.

Genes were also found in the *B. siamensis* YB-1631 genome for different factors related to plant–microbe interactions, utilization of plant-derived substrates, and response to environmental stress. Genes in the *B. siamensis* YB-1631 genome related to plant–microbe interactions were those for chemotaxis, such as chemotaxis protein and methyl-accepting chemotaxis protein [94], motility, such as flagellar motor protein and swarming motility protein [94], colonization and biofilm, such as levansucrase, surface adhesion protein, γ-PGA, extracellular polysaccharide, and biofilm-surface layer protein [94,97], and ISR elicitors, such as elongation factor, flagellin HAP, flagellin HAG, teichuronic acid, and lipopoly-saccharide biosynthesis [94]. Genes for utilization of plant-derived substrates were also found, such as xylanase, chitosanase, bacillopeptidase F, and ABC sugar transporter [94], as were genes potentially important for response to environmental stress, such as cold shock protein and superoxide dismutase [94]. Compared to the genomes of *B. siamensis* KCTC 13613T, *B. siamensis* SCSIO 05746, *B. siamensis* JFL15, and *B. siamensis* RGM 2529, there were the same number of genes for these traits, except for the *B. siamensis* RGM 2529 genome, which was missing the colonization and biofilm formation genes *lytA*, *lytB*, *lytC,* and *yhcK*, and the colonization and biofilm formation gene *pznL* only found in the *B. siamensis* JFL15 genome (Appendix A). This indicates that *B. siamensis* YB-1631 has similar traits for plant–microbe interactions, utilization of plant-derived substrates, and response to environmental stress as other *B. siamensis* isolates, and perhaps *B. siamensis* JFL15 may be less effective in colonizing plant roots. However, future work would need to compare the strains directly for these traits to determine if the differences in the genomes have a significant effect on its biology.

## 5. Conclusions

In this study, a new strain of *B. siamensis*, YB-1631, was examined with strong in vitro antagonism to *F. pseudograminearum*, control of FCR, and plant growth promotion. An examination of different media to evaluate in vitro antifungal production by *B. siamensis* YB-1631 revealed that a low C/N ratio may be a critical factor for production. Future work should use completely defined media where several C/N ratios can be evaluated to choose the optimal one. The complete genome sequence of *B. siamensis* YB-1631 reveals many genes potentially related to biocontrol, plant growth promotion, and ISR. While similar to the genes found in the genomes of other *B. siamensis* biocontrol isolates, there were some differences. Sequencing more genomes of *B. siamensis* biocontrol isolates should help to reveal the range of genes that may be involved. However, future work should examine each potential biocontrol and plant growth promotion gene to assess their contribution to those traits. Although *B. siamensis* has been shown to be a BCA for a variety of plant diseases, this is the first report on the biocontrol of wheat FCR using a *B. siamensis* isolate.

## Figures and Tables

**Figure 1 jof-09-00547-f001:**
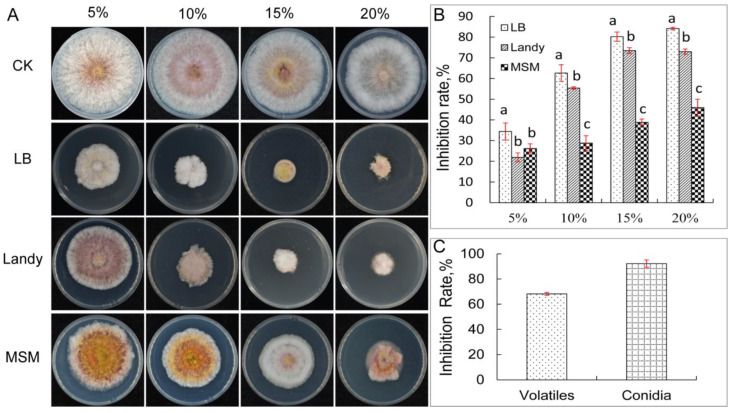
The in vitro antifungal activity of YB-1631 against *F. pseudograminearum*. (**A**) Colonies of *F. pseudograminearum* on PDA with 5%, 10%, 15%, and 20% of cell-free culture filtrates of YB-1631 grown in LB, Landy, or MSM agar. (**B**) Inhibitory rate of *F. pseudograminearum* growth with YB-1631 culture filtrates. (**C**) Inhibitory rate of *F. pseudograminearum* growth with YB-1631 volatile substances and inhibitory rate of *F. pseudograminearum* conidial germination with 66% cell-free culture filtrates of YB-1631 grown in LB. The error bar represents standard deviation (SD), and different letters on the columns in Figure 1B indicate significant differences (*p* < 0.05) among the concentrations of each culture medium using one-way ANOVA with Tukey’s test.

**Figure 2 jof-09-00547-f002:**
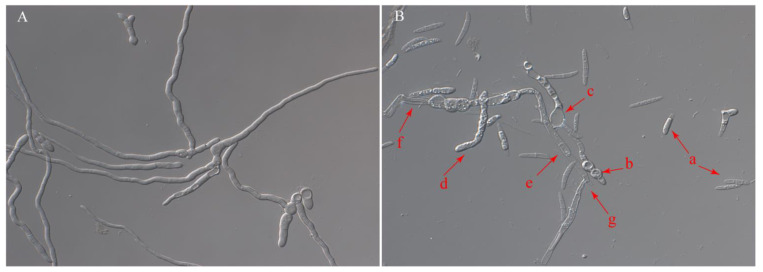
Effect of culture filtrate of YB-1631 on conidial germination of *F. pseudograminearum*. (**A**) Morphology of conidia germinating for 12 h in the control treatment. (**B**) Morphology of conidia germinating for 12 h in 66% LB culture filtrate of YB-1631. Arrows indicate morphological changes in the fungus. Conidia showed lysis (a) and deformation (b), germ tubes showed swelling (c), protoplast condensation (d), shortening of septum intervals (e), wrinkling (f), and lysis (g). The photos were taken with 400× magnifications.

**Figure 3 jof-09-00547-f003:**
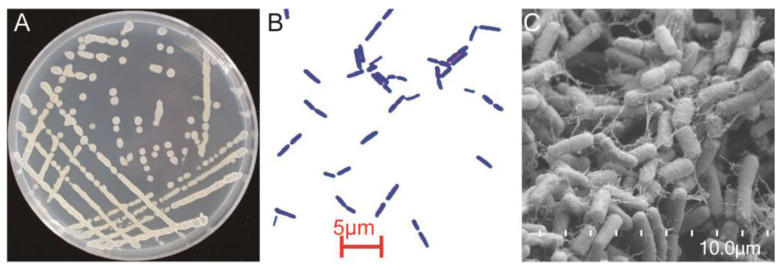
Morphology of strain YB-1631 in culture. (**A**) Colony morphology of YB-1631 on LB agar at 12 h, (**B**) Gram stain, (**C**) morphology under scanning electron microscope.

**Figure 4 jof-09-00547-f004:**
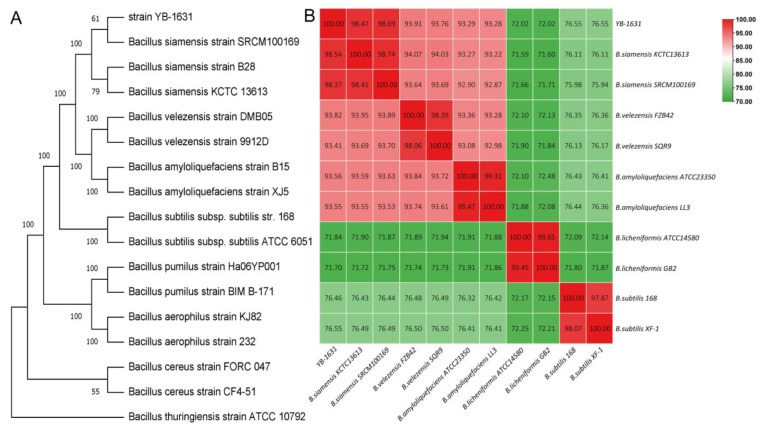
Identification of YB-1631 (**A**) NJ tree of *gyrB* sequences of YB-1631 and strains of *B. siamensis*, *B. velezensis*, *B. amyloliquefaciens*, *B. subtilis*, *B. pumilus*, *B. aerophilus*, *B. cereus,* and *B. thuringiensis*, (**B**) heatmap of ANI values based on complete genome sequences of YB-1631 and strains of *B. siamensis*, *B. velezensis*, *B. amyloliquefaciens*, *B. licheniformis,* and *B. subtilis*.

**Figure 5 jof-09-00547-f005:**
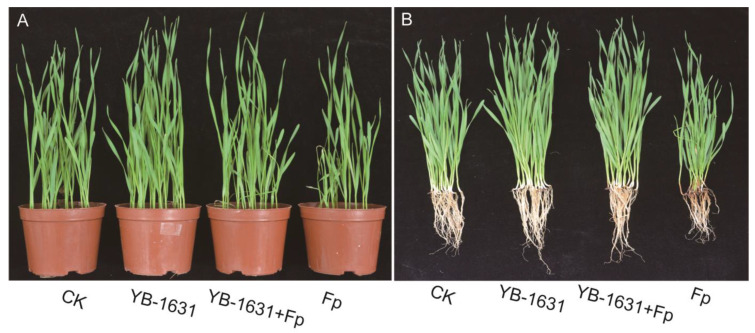
Effect of strain YB-1631 on 21-day-old wheat seedlings with and without *F. pseudograminearum* inoculation. (**A**) Potted seedlings at 21 days after sowing. (**B**) Seedlings with soil removed. CK = sowing seeds with sdH_2_O treatment into sterile soil treatment, YB-1631 = sowing seeds with 10^8^ cells/mL YB-1631 treatment into sterile soil treatment, YB-1631 + Fp = sowing seeds with 10^8^ cells/mL YB-1631 treatment into sterile soil with 2% *F. pseudograminearum* soil inoculum treatment, and Fp = sowing seeds with sdH_2_O treatment into sterile soil with 2% *F. pseudograminearum* soil inoculum treatment.

**Figure 6 jof-09-00547-f006:**
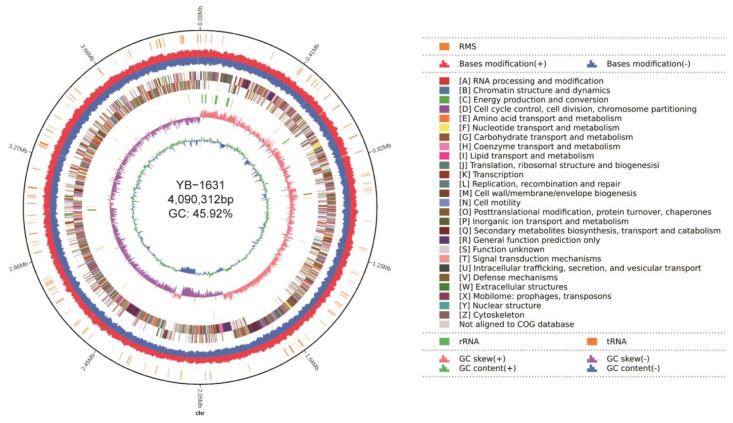
Circular map of *Bacillus siamensis* YB-1631 genome. Circles from the outside to the inside are as follows: ring 1 for genome size (black line), ring 2 for restriction-modification system (RMS), ring 3 for bases modification, forward strand (red), and reverse strand (blue), ring 4 for COG classifications of protein-coding genes on the forward strand and reverse strand, ring 5 for the distribution of tRNAs (brown) and rRNAs (green), ring6 for GC skew, ring 7 for GC content.

**Figure 7 jof-09-00547-f007:**
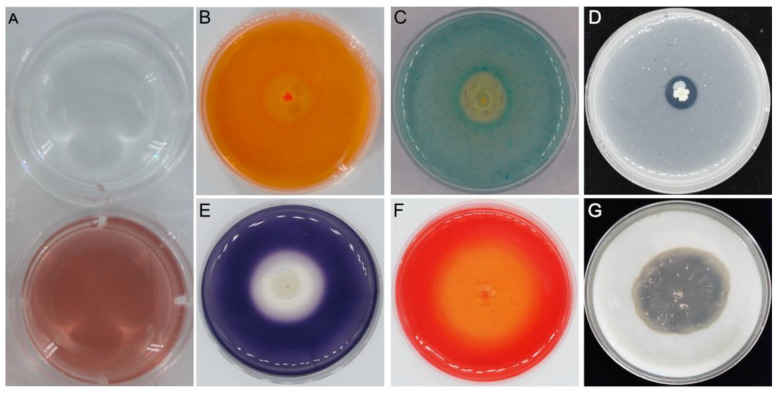
Plant growth promotion and biocontrol traits of strain YB-1631. (**A**) IAA, the upper plate is the non-inoculated broth control, the lower plate is the YB-1631 culture broth, where the broth was placed in a hole in the center of the plate, and the pink color of the broth indicated the presence of IAA. Activities of (**B**) β-1, 3-glucanase, (**C**) siderophores, (**D**) phosphorus solubilization, (**E**) amylase, (**F**) cellulase, and (**G**) protease were visible as transparent circles or light-colored areas around the YB-1631 colony in the center of each plate.

**Table 1 jof-09-00547-t001:** The width of no fungal growth between the *F. pseudograminearum* and soil bacterial colonies on PDA.

Strain	Width	Strain	Width
Mq202003-1	2.33 ± 0.29 lm	Mq202003-30	4.60 ± 0.17 abcde
Mq202003-2	1.17 ± 0.29 no	Mq202003-31	3.40 ± 0.17 hi
Mq202003-3	4.17 ± 0.29 cdefg	Mq202003-32	4.07 ± 0.12 cdefg
Mq202003-4	2.83 ± 0.29 jk	Mq202003-33	3.87 ± 0.12 fgh
Mq202003-5	0.67 ± 0.29 opq	Mq202003-34	2.07 ± 0.12 m
Mq202003-8	0.40 ± 0.17 pq	Mq202003-36	4.40 ± 0.17 bcdef
Mq202003-9	4.83 ± 0.29 ab	Mq202003-37	4.07 ± 0.12 defg
Mq202003-10	4.67 ± 0.29 abc	Mq202003-38	1.23 ± 0.25 no
Mq202003-11	4.00 ± 1.00 fg	Mq202003-39	2.40 ± 0.17 lm
Mq202003-12	4.17 ± 0.29 cdefg	Mq202003-40	3.67 ± 0.29 ghi
Mq202003-13	0.83 ± 0.29 nop	Mq202003-41	3.93 ± 0.12 fg
Mq202003-14	1.00 ± 0.00 no	Mq202003-42	3.13 ± 0.12 ijk
Mq202003-15	2.50 ± 0.50 lm	Mq202003-43	2.33 ± 0.29 lm
Mq202003-16	5.07 ± 0.12 a	Mq202003-44	4.17 ± 0.29 cdefg
Mq202003-17	4.23 ± 0.25 cdefg	Mq202003-45	4.67 ± 0.29 ab
Mq202003-18	4.17 ± 0.29 cdefg	Mq202003-47	4.40 ± 0.17 bcdef
Mq202003-19	3.17 ± 0.29 ijk	Mq202003-48	1.33 ± 0.29 n
Mq202003-20	2.50 ± 0.50 lm	Mq202003-49	3.33 ± 0.29 ij
Mq202003-21	4.17 ± 0.29 cdefg	Mq202003-50	2.17 ± 0.29 lm
Mq202003-22	0.83 ± 0.29 nop	Mq202003-51	2.67 ± 0.29 kl
Mq202003-23	4.67 ± 0.29 abc	Mq202003-53	0.83 ± 0.29 nop
Mq202003-24	3.17 ± 0.29 ijk	Mq202003-54	0.67 ± 0.29 opq
Mq202003-25	2.50 ± 0.50 lm	Mq202003-56	1.33 ± 0.29 n
Mq202003-27	0.23 ± 0.25 q	Mq202003-57	1.40 ± 0.17 n
Mq202003-28	0.67 ± 0.29 opq	Mq202003-58	4.83 ± 0.29 ab

Data in the table are mean ± standard deviation (SD). Means with a different letter (a–q) after the antifungal width indicates significant difference (*p* < 0.05) using one-way ANOVA with Tukey’s test.

**Table 2 jof-09-00547-t002:** Effect of YB-1631 on wheat seedling growth and FCR levels. CK indicates dsH_2_O-treated seeds and no *F*. *pseudograminearum* inoculation, YB-1631 indicates YB-1631-treated seeds and no *F*. *pseudograminearum* inoculation, *Fp* indicates dsH_2_O-treated seeds and *F*. *pseudograminearum* inoculation, YB-1631 + *Fp* indicates YB-1631-treated seeds and *F*. *pseudograminearum* inoculation.

Treatment	Root Length (cm)	Plant Height (cm)	Root Fresh Weight (mg)	Total Fresh Weight (mg)	FCR Incidence (FCRI %)	Disease Severity Index (DSI %)	Relative Control Effect (RCE %)
CK	8.89 ± 0.15 b	25.29 ± 0.58 b	26.36 ± 1.13 c	407.40 ± 6.34 c			
YB-1631	12.19 ± 0.52 a	28.90 ± 0.73 a	31.88 ± 0.58 a	449.60 ± 10.48 a			
*Fp*	6.41 ± 0.19 c	18.15 ± 1.63 c	13.80 ± 2.56 d	225.40 ± 15.87 d	90.75 ± 3.67 a	48.69 ± 1.33 a	
YB-1631 + *Fp*	12.07 ± 0.18 a	27.98 ± 0.40 b	29.80 ± 0.84 b	426.20 ± 5.84 b	14.50 ± 1.60 b	8.17 ± 0.93 b	83.23 ± 1.75

Data in the table are mean ± standard deviation (SD). Means with a different letter (a–d) in each column indicate significant difference (*p* < 0.05) between treatments using one-way ANOVA with Tukey’s test.

**Table 3 jof-09-00547-t003:** Activities of polyphenol oxidase (PPO), catalase (CAT), phenylalanine ammonia-lyase (PAL), superoxide dismutase (SOD), peroxidase (POD), lipoxygenase (LOX), and MDA content in treated wheat seedlings. Treatments were water (CK), bacteria (YB-1631), *F*. *pseudograminearum* inoculation (*Fp*), YB-1631, and *F*. *pseudograminearum* inoculation (YB-1631 + *Fp*).

Treatment	PPO (U/g)	CAT (U/g)	PAL (U/g)	SOD (U/g)	POD (U/g)	LOX (U/g)	MDA (nmol/g)
CK	22.38 ± 1.04 d	143.56 ± 3.11 d	24.86 ± 2.38 c	9.61 ± 1.54 c	15,175.45 ± 249.40 d	752.40 ± 13.14 d	21.36 ± 0.79 b
YB-1631	25.5 ± 0.67 c	156.03 ± 5.19 c	34.17 ± 2.38 b	24.64 ± 1.70 b	16,674.29 ± 384.66 c	3261.24 ± 94.44 c	16.66 ± 2.15 c
*Fp*	38.06 ± 0.46 b	238.68 ± 3.48 b	37.20 ± 1.37 b	31.66 ± 1.64 a	34,486.16 ± 544.57 b	1008.90 ± 22.57 b	37.88 ± 1.52 a
YB-1631 + *Fp*	45.89 ± 1.28 a	567.73 ± 10.22 a	51.76 ± 0.84 a	34.53 ± 0.70 a	42,529.88 ± 651.12 a	4022.64 ± 77.42 a	20.93 ± 2.44 c

Data in the table are mean ± standard deviation (SD). Means with a different letter (a–d) in each column indicate significant difference (*p* < 0.05) between treatments using one-way ANOVA with Tukey’s test.

**Table 4 jof-09-00547-t004:** Putative gene clusters for synthesis of secondary metabolites in *B. siamensis* YB-1631 genome analyzed by antiSMASH.

Clusters	Types	From	To	Most Similar Known Clusters	Similarity
Cluster 1	NRPS	337,277	401,867	Surfactin biosynthetic gene cluster from *B. velezensis* FZB42	82%
Cluster 2	NRPS	1,865,326	1,973,480	Fengycin biosynthetic gene cluster from *B. velezensis* FZB42	93%
Cluster 3	NRPS	3,131,564	3,183,345	Bacillibactin biosynthetic gene cluster from *B. subtilis* subsp. *subtilis* str. 168	100%
Cluster 4	TransAT-PKS	1,709,598	1,809,529	Bacillaene biosynthetic gene cluster from *B. velezensis* FZB42	92%
Cluster 5	TransAT-PKS	2,383,511	2,477,284	Difficidin biosynthetic gene cluster from *B. velezensis* FZB42	100%
Cluster 6	PKS-like	1,011,800	1,053,044	Butirosin A biosynthetic gene cluster from *B. circulans*	7%
Cluster 7	terpene	1,137,529	1,154,562	None	
Cluster 8	terpene	2,002,946	2,023,869	None	
Cluster 9	NRP + Polyketide	1,882,031	1,926,708	Bacillomycin D biosynthetic gene cluster from *B. velezensis* FZB42	84%
Cluster 10	NRP + Polyketide	199,769	267,925	Locillomycin biosynthetic gene cluster from *B. subtilis*	21%

## Data Availability

The genome sequence data and assemblies of *Bacillus siamensis* YB-1631 reported in this paper are associated with the NCBI BioProject: PRJNA894992, BioSample: SAMN31484458. Accession Numbers: CP110268.1 in GenBank. The gyrB sequence data are available at https://www.ncbi.nlm.nih.gov/protein/2325799032 (accessed on 6 November 2022).

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
