# Peer review of "Isolation and Genome-Based Characterization of Biocontrol Potential of Bacillus siamensis YB-1631 against Wheat Crown Rot Caused by Fusarium pseudograminearum"

_jof, 2023, doi:10.3390/jof9050547_

Round 1
Reviewer 1 Report
I think the manuscript is interesting and that many researchers in the field may be interested to take a look on. This paper describes the genome sequence of a native to China Bacillus siamensis strain and its antifungal activity in vitro and in vivo, and growth promotion ability on wheat seedlings. The manuscript is well-written; clear, precise, and easy to understand. I think the manuscript may be published in the present form. However, I think the manuscript falls outside the scope of the journal (just my opinion). The work focus on a bacterium and not on a fungus.
Minor coments.
Page 3. Line 104. Use g instead of rpm or add name of the centrifuge and rotor used.
Author Response
Thank you very much for your comments on our work. However, we feel that the manuscript is in line with this journal”. Firstly, similar articles can be found in the Journal of Fungi, such as: < Kim J A, Song J S, Kim P I, et al. Bacillus velezensis TSA32-1 as a Promising Agent for Biocontrol of Plant Pathogenic Fungi[J]. Journal of Fungi, 2022, 8(10): 1053>,< Tu P W, Chiu J S, Lin C, et al. Evaluation of the antifungal activities of Photorhabdus akhurstii and its secondary metabolites against phytopathogenic Colletotrichum gloeosporioides[J]. Journal of Fungi, 2022, 8(4): 403>, < Al-Mutar D M K, Alzawar N S A, Noman M, et al. Suppression of Fusarium Wilt in Watermelon by Bacillus amyloliquefaciens DHA55 through Extracellular Production of Antifungal Lipopeptides[J]. Journal of Fungi, 2023, 9(3): 336>, and< Essential Oil from Croton blanchetianus Leaves: Anticandidal Potential and Mechanisms of Action. Secondly, this manuscript discussed the effects of biocontrol bacteria on the pathogenic fungus of wheat crown rot disease from the perspective of biocontrol bacteria but shows its effects on the pathogen. Thus, the work is inseparable from the Fusarium pseudograminearum.
Page 3. Line 104. Use g instead of rpm or add name of the centrifuge and rotor used.
We have used g instead of rpm.
Reviewer 2 Report
The manuscript has lots of good quality work and the authors carried out massive studies to investigate the biocontrol potential of B. siamensis YB-1631, especially genome-based characterization. However, the manuscript needs some more elaboration on statistical analysis before its publication.
Author Response
We have added more information about the statistical analysis in section 2.7 and the figure legends.
Reviewer 3 Report
Major comments
Line 19: 58 bacteria isolated, or bacterial isolates or biocontrol agents, please write this sentence.
Line 98: Put a space between 2.5 and cm.
Line 102: Write full name first time before writing abbreviations LB, Landy , or MSM, What is landy and MSM here? Similarly, line 106: PDA, and Line 112 CMC?
Line 114: Write how you diluted conidia, manually or using equipment; write clearly;
Line 116-117: What is 400x? Replications. How? or Microscope magnification?
Line 122: 80 µL of 1 × 108 cells/mL of what? Spores, conidia, or bacterial cells? Write clearly?
Line 134: Here, what is 2% inoculum?
Line 136: What is FCR? Write full names, then abbreviations, when you are writing first time? Also, put a reference for this rating scale?
Line 136: 0 means no symptoms and 7 means what, Is the scale has two values 0 and 7
Line 137: What is the difference between FCR and FCRI? Write clearly
Line 146-154: You have taken samples, 20 days post sowing, for enzyme activities> Why 20 days? Why not 20 days, 30 days, 40 days,? Any references for these activities you have to add?
Line 146-147: Biocontrol agents have volatile compounds and Fusarium is a soil-born pathogen, How do these volatile compounds affect Fusarium in the rhizosphere? Also, biocontrol agent secrete antibiotics as you have shown in the in-vitro experiment. What is the relationship between these enzyme's activities and biocontrol agents' antagonistic activities? The discussion needs improvement.
Line 165: Why only one gene has been used for sequencing [ A 1200 bp gyrB sequence]? Why not other genes, 16S, ? To construct more accurate phylogenetic tree, are more genes required? Any Accession number records in NCBI? Please mention
Line 160-195: You have just grown YB-1631 in LB broth for 12 hours and did genomic sequencing; There is no link between biocontrol work and this whole genome sequencing. Rather than doing whole genomic sequencing, it was better to do interaction studies among strain, wheat and Fusarium; then it was easy to justify that these genes are involved in the growth promotion or antagonistic activities. This work does not seem to be part of biocontrol work. How do you justify it?
Figure 1B and C: Figure C, bars color are almost similar, change color, Figure B and C, donot have control, control (ck) are missing? In Figure C, what do you mean by conidia? CONIDIA IS A TREATMENT OR CONTROL
Figure 1A: In MSM and Landy, biocontrol agents are ineffective; it indicates they are not working in different growing conditions.
Figure 2A and B: They are the same; morphological changes are not visible; if possible, label the biocontrol agents and pathogen with different fluorescent proteins, and monitor their activities. Or After treatment, do TEM or SEM to show morphological changes in the Fusarium
Figure 3C: Threads are extracellular matrix?
Line 249-250: 95%-96% identity threshold for defining bacterial species. Any reference for this? If yes add this threshold in the materials and methods
Figure 4: The biocontrol agent also seem to be similar to B. subtilis and other bacterial species, How do you justify that this is B. siamensis by sequencing only one gyr gene?
Figure 5 and Table 2: I do not see any significant growth in these treatments YB-1631 and YB-1631+Fp . But in Table 2, you have shown a significant difference. Please check carefully.
Table 3: PPO, CAT, PAL, SOD, POD and MDA, the Fp treatment has more activities than the remaining treatments? Why?
Line 309-310: Where is the link for the accession number in NCBI?
Figure 6: We know from the figure, or generally, that bacterial genomes have protein-coding genes, tRNAs, rRNAs, and other potential RNA; what is the importance of this information here in your work? Is there any relationship between these genes with your biocontrol work? Write clearly in the discussion
Table 4: You have mentioned many secondary metabolites, For example, surfactin, fengycin, etc; all these secondary metabolites will work against Fusarium? Or specific metabolites? Justify by doing additional analysis that these secondary metabolites are working against Fusarium.
Figure 7: This figure is not clear, I don't see any control in the holes, and even holes are not visible. Zones are not clear.
Round 2
Reviewer 3 Report
You have incorporated my suggestions, One major concern still remain
you are growing bacterial strains in LB for 24 hours, then sending bacteria for WGS, and getting results you are claiming that these genes are involved in the biocontrol activity. This matter needs to be carefully handled in the future. The experiment should be designed carefully and not waste research funds WGS, when it has no role in your study or you do not know how to set WGS data according to your research.
Author Response
Q: you are growing bacterial strains in LB for 24 hours, then sending bacteria for WGS, and getting results you are claiming that these genes are involved in the biocontrol activity. This matter needs to be carefully handled in the future. The experiment should be designed carefully and not waste research funds WGS, when it has no role in your study or you do not know how to set WGS data according to your research.
A: Firstly, strain YB-1631 was selected as a good biocontrol strain, and its mechanism of disease prevention and growth promotion will be more deeply analyzed. Genome sequencing of strain YB-1631, on the one hand, can be more acaccurate to identify the strain at the species level based on ANI (Average Nucleotide Identity) analysis compared to phylogenetic tree of one to three conserved genes, and to preliminarily evaluate whether the strain has related functional genes related to growth promotion and disease prevention from the genome level. On the other hand, the assembly of strain YB-1631 genome lays the foundation for a more accurate mapping of transcriptome data and gene expression quantification for deciphering biocontrol mechanism among strain YB-1631, pathogens and crops nextly because Bacillus genomes had a lot of variability between two different genomes (other colleagues in our laboratory had done the related work).